# Quantitative Alterations in Short-Chain Fatty Acids in Inflammatory Bowel Disease: A Systematic Review and Meta-Analysis

**DOI:** 10.3390/biom15071017

**Published:** 2025-07-15

**Authors:** Laura Chulenbayeva, Zharkyn Jarmukhanov, Karlygash Kaliyekova, Samat Kozhakhmetov, Almagul Kushugulova

**Affiliations:** 1Laboratory of Microbiome, National Laboratory Astana, Nazarbayev University, Astana 010000, Kazakhstan; zharkyn.jarmukhanov@nu.edu.kz (Z.J.); skozhakhmetov@nu.edu.kz (S.K.); 2Centre for Host-Microbiome Interactions, King’s College London, London WC2R 2LS, UK; 3Department of Pharmacology, Medical University Astana, Astana 010000, Kazakhstan; kaliyekova.k@amu.kz

**Keywords:** short-chain fatty acid, butyrate, acetate, propionate, inflammatory bowel disease, Crohn’s disease, ulcerative colitis

## Abstract

Background: Reduced short-chain fatty acids (SCFAs) in inflammatory bowel disease (IBD) impair the gut barrier and immune function, promoting inflammation and highlighting microbiome-targeted therapies’ therapeutic potential. The purpose of this meta-analysis was to study the changes in SCFAs in IBD and their potential role in the occurrence and development of IBD. Methods: The analysis employed a random-effects model to assess the standardized mean difference (SMD) with a 95% confidence interval. A literature search was conducted in databases from 2014 to 20 July 2024 to identify studies investigating SCFAs in IBD. Results: Subgroup analyses revealed a significant reduction in fecal SCFA levels—specifically butyrate, acetate, and propionate—in all IBD subgroups compared to healthy controls. Active IBD showed a greater decrease in butyrate (*p* = 0.004), and UC showed a notable reduction in propionate (*p* = 0.03). When comparing UC and CD, differences were observed mainly in propionate (SMD = −0.76, *p* = 0.00001). Dietary interventions in IBD patients led to increased SCFA levels, with butyrate showing the most improvement (SMD = 1.03), suggesting the potential therapeutic value of dietary modulation. Conclusions: In conclusion, this meta-analysis demonstrates a significant reduction in fecal SCFA levels in patients with IBD, particularly during active phases of the disease and most markedly in CD.

## 1. Introduction

IBD describes a group of persistent inflammatory disorders affecting the digestive system, chiefly comprising Crohn’s disease (CD) and ulcerative colitis (UC). These conditions develop when genetically predisposed individuals exhibit aberrant immune reactions to gut microbiota [1]. UC presents with uniform inflammatory changes affecting the colon’s mucosal layer, characteristically starting at the rectum and possibly spreading upward to affect remaining portions of the large intestine [2]. CD demonstrates pan-enteric involvement potential, capable of affecting the entire digestive tract from the esophagus to the rectum, with hallmark features including full-thickness bowel wall inflammation and patchy lesions that create a mosaic pattern of diseased and healthy tissue [3]. IBD presents a significant health burden across various regions. In the United States, over 0.7% of the population has been diagnosed with IBD, with peak incidence occurring in early adulthood [4]. In Canada, approximately 322,600 individuals (0.82% of the population) were living with IBD as of 2023, and this prevalence is projected to rise to 1.1% by 2035 [5]. Furthermore, European epidemiological data reveal approximately 1.3 million affected patients, corresponding to 0.2% of the region’s inhabitants [6]. Although IBD predominantly affects young adults, it can occur at any age, with around 25% of cases manifesting before the age of 20 [7].

The specific origins of inflammatory bowel disease remain uncertain, though research indicates it develops through dynamic interactions between inherited susceptibility, immunological abnormalities, and external influences. Specific genetic backgrounds may predispose individuals to IBD, while a deviation in immune response to the gut microbiota contributes to chronic intestinal inflammation. Environmental influences, such as diet, smoking, and antibiotic use, have also been implicated in disease onset and progression [8,9,10,11]. Notably, emerging evidence links an elevated intake of highly processed food products with an increased risk of developing IBD [12,13].

Treatment strategies for CD and UC differ significantly due to their distinct pathophysiological and clinical features. While UC is limited to the colon and primarily affects the mucosal layer, CD can involve the entire gastrointestinal tract with transmural inflammation. Consequently, therapeutic approaches vary: aminosalicylates (e.g., mesalamine) are commonly used as a first-line therapy in mild-to-moderate UC but have limited efficacy in CD. In contrast, CD often requires an early initiation of immunosuppressants or biologics (e.g., anti-TNF agents) to control deep tissue inflammation and prevent complications such as strictures or fistulas. Both conditions may benefit from biologic therapies, but treatment goals and drug selection are tailored to disease behavior, location, and severity [14,15].

Recent scientific evidence highlights the crucial role of gut microbiota composition and metabolic function in the development of IBD [16,17]. Studies have consistently reported a reduction in beneficial bacterial taxa such as *Firmicutes*, including *Faecalibacterium prausnitzii*, *Roseburia*, and *Ruminococcus*, along with an enrichment of potentially pathogenic *Proteobacteria*, particularly members of the *Enterobacteriaceae* family [18,19]. In CD, the microbial community shifts toward an increased abundance of *Pasturellaceae*, *Veillonellaceae*, *Neisseriaceae*, *Fusobacteriaceae*, and *Escherichia coli*, while taxa associated with a healthy gut microbiome—such as *Bacteroides*, *Clostridiales*, *Faecalibacterium*, *Roseburia*, *Blautia*, *Ruminococcus*, and *Lachnospiraceae*—are significantly reduced [20]. In ulcerative colitis, there is a notable increase in *Proteobacteria* and *Patescibacteria*, accompanied by a marked decrease in *Desulfobacterota* and *Verrucomicrobiota* compared to healthy individuals [21]. These microbial shifts are not merely secondary to inflammation but actively contribute to the pathogenesis of IBD by disrupting immune regulation, impairing mucosal barrier function, increasing intestinal permeability, altering microbial metabolite production, and sustaining chronic intestinal inflammation [22,23]. Patients with IBD also exhibit a substantial reduction in the abundance of SCFA-producing bacteria.

SCFAs’ essential bacterial byproducts derived from dietary fiber fermentation by intestinal microbiota demonstrate marked quantitative and qualitative changes in IBD that may exacerbate pathological processes [24,25,26]. Among SCFAs, acetate, propionate, and butyrate are the most abundant and play pivotal roles in maintaining intestinal homeostasis. These metabolites exert anti-inflammatory effects, enhance the integrity of the epithelial barrier, regulate immune responses, and serve as an essential energy source for colonocytes [27,28,29]. Understanding the alterations in SCFA production and their metabolic outputs is vital for elucidating the pathophysiology of IBD and for the development of microbiome-based therapeutic strategies.

## 2. Methods

### 2.1. Literature Search Strategy and Study Selection

The literature search was conducted using the English-language databases PubMed and Embase. The search covered the period from 1 January 2014 to 20 July 2024. The following search terms were used: “inflammatory bowel disease,” “IBD”, “Crohn’s disease”, “CD”, “ulcerative colitis”, “UC”, “short-chain fatty acid”, “SCFA”, “acetate”, “acetic acid”, “propionate”, “propionic acid”, “butyrate”, and “butyric acid.” Boolean operators (AND, OR, NOT) were applied to broaden or narrow the search results. To ensure comprehensiveness, the search was further refined by manually screening the reference lists of relevant original articles. The literature search and study selection were conducted according to the steps outlined in this section and were in accordance with the Preferred Reporting Items for Systematic Reviews and Meta-Analyses (PRISMA) guidelines. The research protocol was registered with “Quantitative Alterations in Short-Chain Fatty Acids in Inflammatory Bowel Disease: A Systematic Review and Meta-Analysis”, and the registration number is CRD420251039646. Language restrictions were applied.

### 2.2. Selection of Studies Involving Measurement of SCFA Concentrations in Stool

In the present study, the published literature was extracted from the aforementioned databases, specifically focusing on the assessment of SCFA concentrations in IBD. This included studies that investigated the modulation of SCFA profiles and/or disease status in patients with CD and UC, aged 18 years and older, where such data were available. To be included in the core dataset, studies had to involve patients diagnosed with IBD and include an analysis of fecal SCFA levels. Only trials that reported actual quantitative values of SCFA were considered for analysis. The inclusion and exclusion criteria were as follows:

Inclusion criteria: Studies evaluating a quantitative analysis of SCFA concentrations in the stool samples of healthy subjects and patients diagnosed with CD or UC; studies involving patients with CD or UC before and after dietary interventions.

Exclusion criteria: Studies that did not report any SCFA measurements; studies involving SCFA analysis in samples other than stool; studies involving participants under the age of 18.

### 2.3. Data Extraction

Study data were abstracted independently by multiple investigators for each eligible publication, including the (1) first author’s name and year of publication; (2) country of origin; (3) study design; (4) type of IBD (UC or CD); (5) case and control groups; (6) number of cases, age, and sex; (7) unit of SCFA measurement; (8) data format provided for SCFA; (9) methods used for SCFA analysis; (10) units of SCFA measurement; and (11) SCFA measurement methods, which were treated as continuous variables. Any discrepancies were resolved through discussion with the academic supervisor.

### 2.4. Statistical Analysis

All statistical analyses were conducted in Review Manager 5.4.1 (RevMan). Standardized mean differences (SMDs) and 95% CIs were computed employing a random-effects approach. Study heterogeneity was evaluated through I^2^ (quantifying variability percentage), chi-squared tests, and Tau^2^ (estimating between-study effect size variance). Overall effect significance was determined via Z-tests (*p* < 0.05 threshold), with risk ratios (RR) and 95% CIs serving as primary effect measures. A forest plot was generated to visualize the results. Sensitivity analysis indicated that the findings were robust to the exclusion of individual studies.

## 3. Results

### 3.1. Literature Search and Study Selection

A total of 2124 records were initially retrieved from database searches using targeted keywords. Following the elimination of duplicates and studies not aligned with the research focus, 315 titles and abstracts were screened. Of these, 301 were excluded for reasons such as irrelevance to the study objective, single-arm design, case reports or series, animal-based research, and review articles. Subsequently, the full texts of 13 studies were evaluated for eligibility, resulting in the inclusion of nine studies that fulfilled all predefined criteria for the meta-analysis [30,31,32,33,34,35,36,37,38] (Figure 1).

### 3.2. Study Characteristics

Table 1, Table 2 and Table 3 present the characteristics of each included study. The selected studies were published between 2018 and 2023 and comprised one prospective study, one randomized controlled trial, and six case–control studies. Geographically, five studies were conducted in Poland, while the remaining four were carried out in Turkey, Japan, the Netherlands, and Australia. All included studies matched appropriate age and sex criteria and assessed SCFAs in fecal samples collected from patients with IBD. Regarding IBD subtypes, five studies included patients with both UC and CD, two studies focused exclusively on UC patients, and one study did not specify the IBD subtype. In terms of SCFA detection methods, capillary electrophoresis or capillary electrophoresis combined with spectrophotometry (CE-U) were commonly used across the included studies. Other methods utilized included gas chromatography, gas chromatography–mass spectrometry (GC-MS), liquid chromatography–tandem mass spectrometry (LC-MS/MS), and high-performance liquid chromatography (HPLC).

### 3.3. Meta-Analyses

The present study explored evaluating the alterations in the levels of butyrate, acetate, and propionate among patients with IBD, stratified by disease activity and type.

Figure 2, Figure 3 and Figure 4 present the meta-analysis of butyrate, acetate, and propionate concentrations (µg/g) in IBD subgroups compared to healthy controls. The results are subdivided based on disease activity (active/inactive) and IBD subtype CD and UC. SMDs with 95% confidence intervals (CIs) are displayed at the study level and in aggregations for each subgroup and the overall analysis.

According to the analysis (Figure 2), in patients with active IBD, the aggregated effect size indicated significantly lower levels of diet compared to healthy controls (SMD = −2.18, 95% CI: [−3.67, −0.68], *p* = 0.004). In contrast, no significant difference was observed between patients with inactive IBD and healthy controls (SMD = −0.57, 95% CI: [−1.99, 0.85], *p* = 0.43). Patients with CD had lower butyrate levels compared to controls, although the difference did not reach statistical significance (SMD = −1.44, 95% CI: [−2.92, 0.04], *p* = 0.06). A trend toward reduced butyrate levels was also observed in UC patients (SMD = −1.07, 95% CI: [−2.34, −0.19], *p* = 0.10), though this was also not statistically significant.

In the overall analysis, butyrate levels were significantly lower in patients with IBD compared to healthy controls across all subgroups (SMD = −1.37, 95% CI: [−1.93, −0.81], *p* < 0.0001). Significant reductions were observed in patients with active IBD (*p* = 0.004) and in the total pooled analysis (*p* < 0.0001). However, no significant reductions were observed in the inactive IBD, CD, and UC subgroups, although the CD subgroup demonstrated a borderline effect (*p* = 0.06). High heterogeneity was present across all subgroups (I^2^ > 80%), suggesting substantial variability among studies. These findings indicate a significant depletion of butyrate in IBD, particularly during active disease phases, underscoring the prospective role of butyrate in intestinal homeostasis and inflammatory processes.

In patients with IBD, acetate levels were significantly lower compared to healthy individuals (*p* < 0.0001) (Figure 3). This reduction was evident across all IBD subtypes. Specifically, patients with active IBD exhibited markedly reduced acetate levels (SMD = −1.52, 95% CI: [−2.56, −0.48], *p* = 0.004), as did those with CD (SMD = −0.75, 95% CI: [−1.45, −0.05], *p* = 0.04), and ulcerative colitis (SMD = −1.12, 95% CI: [−2.09, −0.16], *p* = 0.02), all showing high heterogeneity (I^2^ > 80%), suggesting substantial variability across studies.

Moreover, even in patients with inactive IBD, acetate levels remained significantly lower compared to controls (SMD = −0.88, 95% CI: [−1.73, −0.03], *p* = 0.04). The moderate heterogeneity in this subgroup (I^2^ = 60%) indicates relatively consistent results across studies. This suggests that while acetate depletion persists during remission, it is less pronounced than during active inflammation. The more substantial depletion observed in active IBD (I^2^ = 84%) points to variability in inflammatory status across studies.

These findings imply that acetate deficiency may be implicated in the underlying pathology of IBD. The overall effect size (SMD = −1.07, 95% CI: [−1.45, −0.69]) confirms a strong association between reduced acetate levels and IBD. The high overall heterogeneity (I^2^ = 85%) reflects considerable variability between studies, likely attributable to differences in study design, patient populations, and disease severity. However, subgroup analysis (Chi^2^ = 1.57, *p* = 0.67, I^2^ = 0%) indicates no significant differences between the subgroups (i.e., active IBD, inactive IBD, CD, and UC), suggesting a consistent trend of acetate reduction across disease types and states.

The compiled SMD for propionate in the active IBD subgroup was −1.24 (95% CI: [−2.53, 0.05]), indicating a trend toward lower propionate levels in patients with active IBD compared to healthy controls, although the results were not statistically significant (*p* = 0.06). Heterogeneity was high (I^2^ = 90%, *p* < 0.0001), pointing to substantial inconsistencies in study outcomes. This modest effect implies that while propionate levels may be reduced in active IBD, additional data are needed to confirm this trend (Figure 4).

In the inactive IBD (SMD = −0.16, 95% CI: [−1.36, 1.03]) and CD (SMD = −0.33, 95% CI: [−1.09, 0.43]) subgroups, the pooled effect sizes did not demonstrate significant differences in propionate levels compared to healthy controls (*p* = 0.79 and *p* = 0.39, respectively). Moderate heterogeneity was observed in the active IBD subgroup (I^2^ = 74%, *p* = 0.02), indicating some inconsistency across studies, while high heterogeneity in the CD subgroup (I^2^ = 84%, *p* < 0.0001) suggests study-specific differences that warrant further investigation.

For patients with UC, the effect size (SMD = −1.04, 95% CI: [−1.98, −0.09]) revealed a significant reduction in propionate levels compared to controls (*p* = 0.03). However, the high heterogeneity (I^2^ = 88%, *p* < 0.0001) indicates considerable variability across studies, potentially influenced by differences in patient populations, methodologies, or disease severity. This finding suggests that propionate metabolism may be particularly affected in UC, potentially due to gut microbiota alterations or disease-associated inflammation.

Overall, propionate levels revealed significantly depressed levels in the IBD group compared to healthy controls (*p* = 0.002). No significant differences were found between subgroups (*p* = 0.43), suggesting a consistent trend of propionate reduction across different disease types. Nevertheless, the very high overall heterogeneity (I^2^ = 91%) reflects substantial between-study variability, which may impact the reliability of these results.

The forest plot in Figure 5 presents a meta-analysis comparing the levels of butyrate, acetate, and propionate between patients with UC and CD. Four studies provided data for this comparison. The SMDs for butyrate, acetate, and propionate were 0.56 (95% CI: [–0.37 to 1.48], *p* = 0.24), 0.30 (95% CI: [–0.65 to 0.05], *p* = 0.09), and –0.76 (95% CI: [–0.95 to –0.57], *p* = 0.00001), respectively. These results indicate a trend toward higher butyrate concentrations in UC patients; however, the meta-analysis did not show a statistically significant difference in butyrate levels between UC and CD. The pooled estimate for acetate suggests a slight, non-significant tendency toward lower acetate levels in UC compared to CD (*p* = 0.09). In contrast, propionate levels were significantly lower in patients with UC than in those with CD. Moderate heterogeneity was observed in the analysis of acetate levels (I^2^ = 34%, *p* = 0.21), suggesting relatively minor variability among studies compared to other analyses. For propionate, the narrow confidence interval and absence of heterogeneity (I^2^ = 0%) further support the robustness of these findings.

We also conducted an analysis of SCFA production in response to dietary intervention in patients with IBD (Figure 6). The SMDs for butyrate, acetate, and propionate levels before and after the dietary intervention were 1.03 (95% CI, 0.52 to 1.54), 1.12 (95% CI, −1.22 to −3.46), and 0.58 (95% CI, −0.26 to 1.43), respectively (Figure 5). The studies included in the analysis of butyrate levels showed an I^2^ value of 0%, indicating no observed heterogeneity among these studies. In contrast, studies assessing acetate and propionate levels exhibited substantial heterogeneity, with I^2^ values exceeding 50%.

## 4. Discussion

IBD, encompassing CD and UC, has become an enduring global healthcare challenge, with incidence rates steadily rising. Economic development triggers societal transformations including urban expansion, industrial growth, agricultural modernization, healthcare system advancement, improved sanitation, and lifestyle modifications. Within this evolving environment, inflammatory bowel diseases establish their presence and proliferate [39,40]. The gut microbiota structure of IBD patients was considerably different from that of healthy persons at different taxonomic levels. Four major bacterial phyla—*Firmicutes*, *Bacteroidetes*, *Proteobacteria*, and *Actinobacteria*—which collectively make up the majority of the gut microbiota were found to have significant imbalances. These bacterial groups coexist poorly with normal commensal gut bacteria in contrast to bacterial groups that remained unchanged in IBD patients [41]. *Actinomyces*, *Veillonella*, and *Escherichia coli* were shown to be more prevalent in CD or UC, but *Christensenellaceae* and *Coriobacteriaceae*, as well as *Faecalibacterium prausnitzii*, were found to be less prevalent. In contrast, there was a decrease in *Eubacterium rectale* and *Akkermansia* and an increase in *E. coli* in patients with UC [42].

The gut microbiome interacts with host cells to produce various microbial metabolites and maintain gastrointestinal balance. One such metabolic product, known as SCFAs, are formed when dietary fiber undergoes fermentation [43,44]. Propionate production via the succinate pathway is primarily carried out by members of the Bacteroidetes and Firmicutes phyla, while the propanediol pathway involves bacteria from the *Lachnospiraceae* family [45,46]. Butyrate is produced by species such as *Bifidobacteria*, *Faecalibacterium prausnitzii*, *Eubacterium rectale*, and *Eubacterium hallii* and *Coprococcus* [47,48]. The primary acetate-producing bacteria include members of the *Bacteroidetes phylum*, such as *Bacteroides* and *Prevotella* species, as well as *Bifidobacterium* species and *Akkermansia muciniphila* [49]. At high concentrations, secreted acetate has been shown to inhibit the growth of *E. coli* [50].

SCFAs, such as acetate, propionate, and butyrate, are essential bacterial byproducts generated when gut microbiota metabolize indigestible carbohydrates in the large intestine. These microbial metabolites are vital for sustaining gut homeostasis and have become a major research focus in IBD investigations. Contemporary research has elucidated multiple pathways through which SCFAs modulate IBD development. They strengthen the gut barrier by increasing tight-junction protein expression and promoting mucin secretion, which limits pathogenic invasion and mitigates IBD-related inflammatory responses [51,52]. These metabolites also interact with G protein-coupled receptors (GPCRs), such as GPR41, GPR43, and GPR109A, leading to anti-inflammatory effects. Furthermore, SCFAs act as histone deacetylase (HDAC) inhibitors, suppressing TLR4 expression, enhancing the differentiation of regulatory T cells (Tregs), and inducing the secretion of anti-inflammatory cytokines such as IL-10, which are essential for mitigating intestinal inflammation in IBD [53]. The presence of SCFAs also influences the diversity and stability of the gut microbiota. Alterations in SCFA concentrations can impact microbial balance, potentially exacerbating or alleviating IBD symptoms [54].

In our study, significantly lower levels of butyrate, acetate, and propionate were observed in patients with IBD, with the reduction particularly associated with active disease. This suggests a potential role for these SCFAs in intestinal inflammation. The meta-analysis demonstrated that the decrease in fecal butyrate concentration was more pronounced in patients with active IBD, indicating a possible link between disease activity and impaired SCFA metabolism, specifically in terms of butyrate production or utilization. Multiple studies and meta-analyses have reported significantly reduced butyrate levels in patients with IBD, especially during the active phases of the disease [55,56,57]. This supports the notion that active inflammation may be closely associated with decreased butyrate production or increased consumption, potentially due to an altered gut microbiota composition or impaired colonic absorption [58,59].

Subgroup analysis revealed a consistent trend toward reduced butyrate levels in both CD and UC, although statistical significance varied. The lack of significance in cases of inactive IBD may suggest partial restoration of microbial balance and SCFA production during remission or may reflect methodological variability and smaller sample sizes in these studies. Elevated butyrate levels observed in UC could indicate a preserved fermentation capacity compared to CD, where gut dysbiosis and inflammation may impair SCFA production [60]. Although the findings are not statistically significant, they highlight the potential role of butyrate in differentiating UC from CD. As demonstrated in the study by Takahashi et al., patients with CD showed a marked reduction in butyrate-producing bacteria, including *Bacteroides*, *Eubacterium*, *Faecalibacterium*, and *Ruminococcus*, contributing to butyrate depletion [61]. Further large-scale, well-controlled studies are needed to clarify this association and to determine whether butyrate supplementation may offer therapeutic benefits, particularly in CD patients with markedly reduced butyrate levels. Butyrate, a key SCFA, serves as a primary energy source for colonocytes, supports gut barrier integrity, and regulates intestinal motility through interactions with SCFA receptors and gut hormones [62]. It enhances water and electrolyte absorption and modulates gene expression as a histone deacetylase (HDAC) inhibitor, contributing to its anti-tumor effects [63,64]. Butyrate also exhibits anti-inflammatory properties by inhibiting NF-κB signaling and cytokine production, influencing both gut and brain function via MCT1 transport [65,66].

The analysis also highlights substantial heterogeneity (I^2^ > 90% in most subgroups), likely attributable to differences in study design, disease classification, dietary patterns, geographic regions, and the analytical methods used to quantify SCFA levels. In the studies included in our meta-analysis, various analytical techniques were employed for the quantification of short-chain fatty acids (SCFAs), including capillary electrophoresis with UV detection (CE-UV), gas chromatography–mass spectrometry (GC-MS), liquid chromatography–tandem mass spectrometry (LC-MS/MS), gas chromatography (GC), and high-performance liquid chromatography (HPLC). This methodological diversity represents a significant source of heterogeneity, particularly due to differences in analytical sensitivity and specificity. For example, CE-UV is generally less sensitive than more advanced techniques such as GC-MS and may be limited in its ability to detect SCFAs at low concentrations [67]. Although GC-MS provides high sensitivity and selectivity, it requires expensive instrumentation, specialized technical expertise, and additional steps such as derivatization, which increase analytical complexity [68]. HPLC, in turn, may exhibit lower sensitivity and resolution compared to GC-MS and LC-MS/MS, particularly when analyzing complex sample matrices [67]. These methodological differences should be carefully considered, as they may significantly contribute to the high levels of heterogeneity (I^2^) observed in the results of this meta-analysis. Despite this variability, the overall effect remains robust, underscoring the clinical relevance of butyrate deficiency in IBD.

This meta-analysis also demonstrates a consistent and significant reduction in fecal acetate levels in patients with IBD compared to healthy controls. The pooled SMDs indicate a moderate-to-large effect size, suggesting that dysbiosis in IBD is associated with altered microbial fermentation activity affecting acetate production. Subgroup analyses based on disease activity and type further support these findings. Patients with active IBD exhibited a more pronounced reduction in acetate levels, although levels were also significantly lower in cases of inactive IBD, albeit with a smaller effect size. Acetate depletion was observed in both CD and UC, with the effect being more marked in CD. However, this was accompanied by high heterogeneity across studies. It is noteworthy that most gut microorganisms do not generate butyrate directly from dietary fiber. Instead, they often depend on intermediary metabolites—primarily acetate and, to a lesser extent, propionate—as critical substrates in the pathways that lead to butyrate synthesis. This interdependence, known as cross-feeding, is a fundamental aspect of metabolic interactions within the intestinal microbial ecosystem and underscores the complex symbiotic relationships between different bacterial groups [69]. For example, *Faecalibacterium prausnitzii*, a key butyrate-producing species in the colon, utilizes acetate in the butyryl–CoA/acetate CoA–transferase pathway to synthesize butyrate [70]. Conversely, only a few species, such as *Clostridium butyricum*, are capable of converting dietary fiber directly into butyrate without relying on such intermediates [45].

Understanding this metabolic interplay is essential for the accurate interpretation of fecal SCFA profiles. A decrease in acetate levels, especially when pronounced, should not automatically be attributed to reduced microbial synthesis alone. It may also indicate a heightened utilization of acetate by butyrate-producing microbes that depend on it for their metabolic activity. In IBD, where butyrate-producing bacterial populations are frequently depleted, such shifts in intermediate metabolites like acetate may reflect underlying disruptions in microbial cross-feeding dynamics and overall metabolic function within the gut microbiota.

The observed reduction in acetate levels may have multiple physiological implications. Acetate, primarily produced by *Bifidobacterium*, *Bacteroides*, and certain *Firmicutes* species, plays a critical contribution in maintaining epithelial barrier integrity, modulating immune responses, and regulating colonic pH to inhibit the growth of pathogenic organisms [71,72,73]. Acetate enhances intestinal barrier function through the upregulation of tight-junction components and the stimulation of mucin secretion, consequently inhibiting microbial penetration and suppressing immune activation in the mucosal lining. Through the activation of immune cell GPR43 receptors, it downregulates the synthesis of pro-inflammatory signaling molecules [74]. In addition, acetate contributes to the differentiation of regulatory T cells (Tregs) and the secretion of the anti-inflammatory cytokine IL-10. Recent studies also suggest that acetate promotes the production of immunoglobulin A (IgA), which selectively targets pathobionts such as *Enterobacterales*, thereby aiding in microbial containment [75].

The high heterogeneity observed across studies (I^2^ = 85%) likely reflects differences in sample processing, quantification methods, dietary patterns, disease severity, and geographic populations. Despite this variability, the direction of the effect remained consistent across all subgroups, supporting the reliability of the findings. According to the conducted meta-analysis, fecal propionate levels were significantly reduced in individuals with IBD compared to healthy controls, as indicated by the SMDs. Among IBD subtypes, the reduction in propionate was most pronounced in patients with UC, while the decrease observed in CD was smaller and not statistically significant. Furthermore, patients with active IBD showed a trend toward lower propionate levels, suggesting that disease activity may further suppress SCFA production. In contrast, no significant difference was observed in individuals with inactive IBD, which may indicate partial microbial recovery during remission.

Propionate also exerts immunomodulatory effects by activating G protein-coupled receptors on immune cells, leading to the downregulation of pro-inflammatory cytokine production [76]. It contributes to the maintenance of the intestinal epithelial barrier by enhancing the expression of tight-junction proteins, thereby reducing intestinal permeability and preventing bacterial translocation [77]. Furthermore, reduced propionate levels limit its beneficial systemic effects, such as lipid regulation and glucose homeostasis, potentially linking IBD to associated metabolic comorbidities [78].

The results of this meta-analysis reveal that the levels of examined SCFAs are more markedly depleted in CD than in UC. Specifically, the analysis provides compelling evidence that propionate levels are significantly lower in UC compared to CD, with highly consistent findings across studies. The absence of heterogeneity (I^2^ = 0%) further supports the reliability of this result. This aligns with previous studies suggesting that SCFA production is more severely impaired in CD due to its direct impact on the colonic mucosa. CD demonstrated the strongest evidence of reduced acetate levels among IBD subtypes, whereas the UC and active IBD groups did not reach statistical significance [79]. CD is associated with more pronounced SCFA depletion than UC, likely due to deeper tissue damage and more severe microbial dysbiosis [80]. The data demonstrate that while diminished butyrate production represents a consistent hallmark of IBD, the degree of depletion appears contingent upon both disease classification and progression stage [81]. Given the pivotal role of butyrate in maintaining intestinal epithelial integrity, reducing inflammation, and regulating immune responses, its deficiency may contribute to the pathogenesis and persistence of the disease [82]. Studies also suggest that genes responsible for propionate synthesis are depleted in CD patients, indicating a diminished capacity for propionate production in this population [83].

Additionally, an evaluation of SCFA levels following dietary therapy was conducted to address the therapeutic gap in UC. Interventions such as the consumption of a fermented beverage [32] and increased intake of fermentable fiber combined with protein restriction [34] had a significant impact on SCFA levels. Notably, fecal butyrate concentrations increased substantially. The low heterogeneity across studies (I^2^ = 0%) suggests a reliable and consistent effect, despite differences in sample sizes and intervention types. While the point estimate indicates a potential increase in acetate levels, high variability among studies renders this result statistically non-significant and inconclusive. Propionate levels showed a modest upward trend; however, the results were not statistically significant and the observed heterogeneity suggests differential effects across studies. Nevertheless, the limited number of available studies highlights the need for larger, well-controlled trials to validate these findings and to investigate their long-term clinical implications.

The findings support the concept that modifying diet or the intestinal environment can enhance the microbial production of SCFAs, which is often diminished in conditions such as IBD or dysbiosis-related disorders [84,85,86]. Although all three SCFAs are reduced in patients with IBD, butyrate exhibits the most pronounced decrease, particularly during active inflammation. Acetate shows a consistent decline across disease subtypes and activity levels, whereas the reduction in propionate appears to be more specific to UC. These results highlight the potential of SCFA profiling as a biomarker for disease activity, subtype differentiation, and a target for therapeutic modulation through dietary strategies or microbiota-directed interventions.

It should be acknowledged that this meta-analysis has certain limitations. Firstly, due to the limited sample sizes in most of the included studies, there is a potential risk of overestimating the intervention effects, a phenomenon commonly observed in smaller trials compared to larger, more robust studies. Secondly, in most of the studies included, the I^2^ statistic exceeded 50%, indicating a moderate level of heterogeneity among the results. This heterogeneity may be attributed to differences in study methodology, sample sizes, diet, medication and the types of medication, and the severity and extent of IBD included in the analysis.

## 5. Conclusions

In conclusion, this meta-analysis demonstrates a significant reduction in fecal SCFA levels in patients with IBD, particularly during active phases of the disease and most markedly in CD. The findings support the hypothesis that disrupted SCFA metabolism is associated with the pathophysiology of IBD. However, the high degree of heterogeneity across studies suggests underlying variability that should be further explored in future research.

## Figures and Tables

**Figure 1 biomolecules-15-01017-f001:**
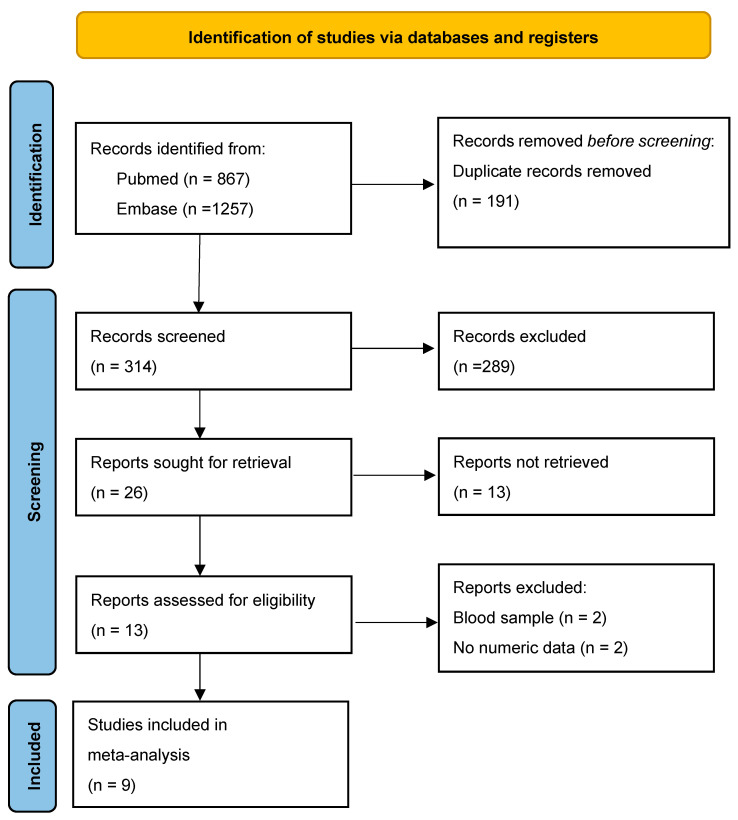
Flowchart of study selection.

**Figure 2 biomolecules-15-01017-f002:**
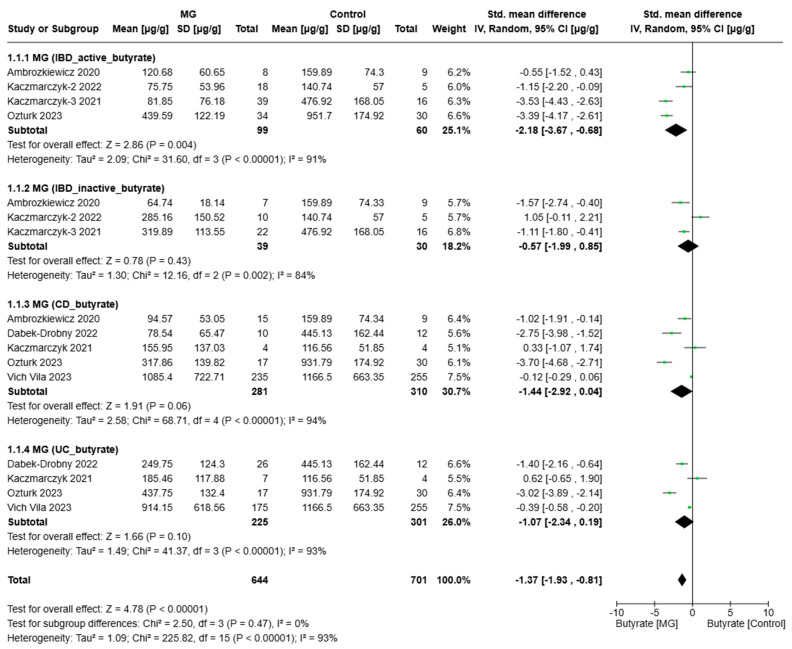
Forest plots of butyrate alterations in MG (IBD patients) vs. Control (Ambrozkiewic 2020 [38]; Kaczmarczyk-2 2022 [31]; Kaczmarczyk-3 2021 [37]; Dabek-Drobny 2022 [36]; Kaczmarczyk 2021 [35]; Ozturk 2023 [33]; Vich Vila 2023 [30]).

**Figure 3 biomolecules-15-01017-f003:**
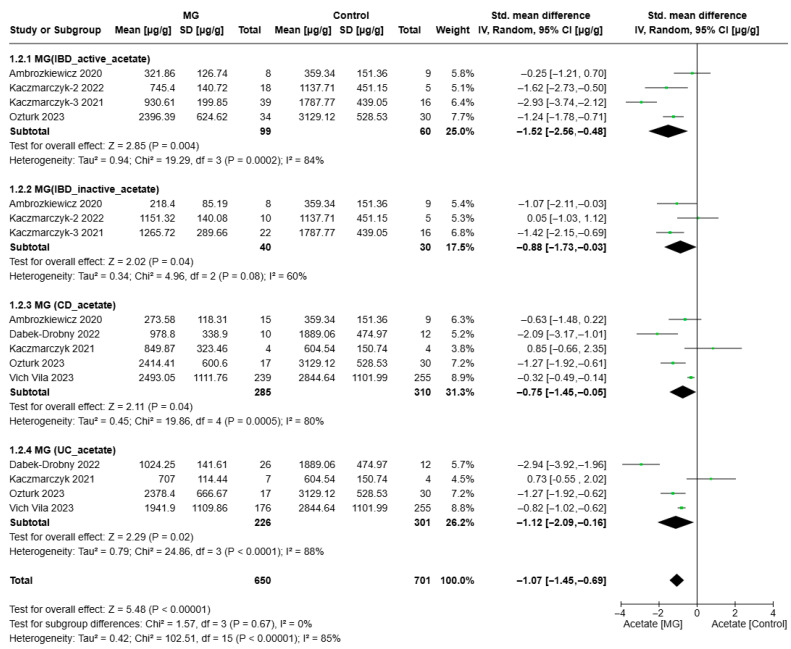
Forest plots of acetate alterations in MG (IBD patients) vs. Control (Ambrozkiewic 2020 [38]; Kaczmarczyk-2 2022 [31]; Kaczmarczyk-3 2021 [37]; Dabek-Drobny 2022 [36]; Kaczmarczyk 2021 [35]; Ozturk 2023 [33]; Vich Vila 2023 [30]).

**Figure 4 biomolecules-15-01017-f004:**
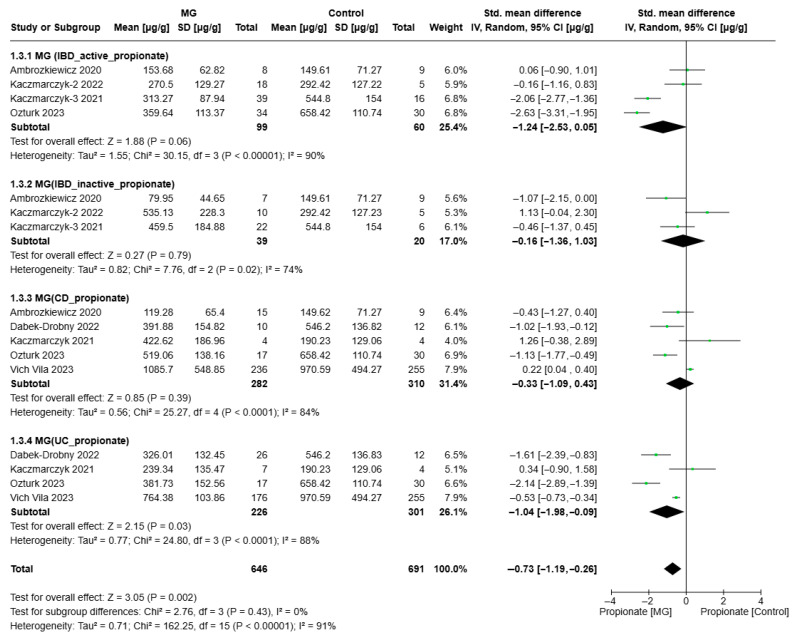
Forest plots of propionate alterations in MG (IBD patients) vs. Control (Ambrozkiewic 2020 [38]; Kaczmarczyk-2 2022 [31]; Kaczmarczyk-3 2021 [37]; Dabek-Drobny 2022 [36]; Kaczmarczyk 2021 [35]; Ozturk 2023 [33]; Vich Vila 2023 [30]).

**Figure 5 biomolecules-15-01017-f005:**
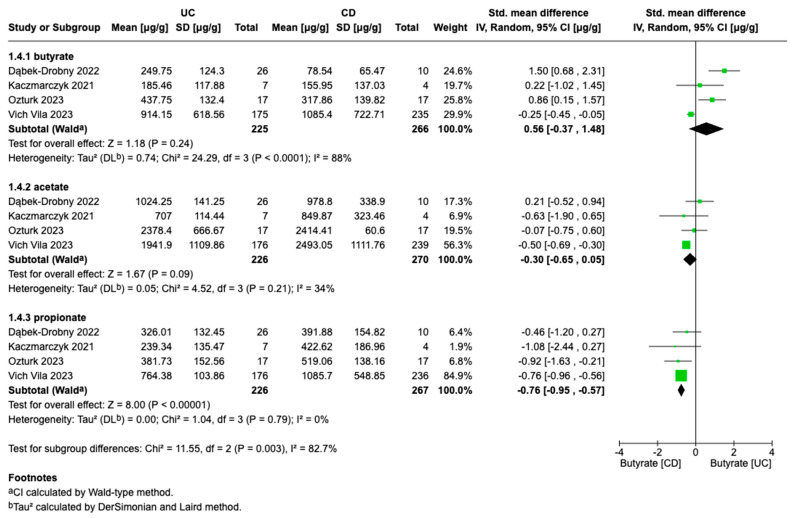
Forest plots of SCFA alterations in UC patients vs. CD patients: 1.4.1 butyrate, 1.4.2 acetate, 1.4.3 propionate (Dabek-Drobny 2022 [36]; Kaczmarczyk 2021 [35]; Ozturk 2023 [33]; Vich Vila 2023 [30]).

**Figure 6 biomolecules-15-01017-f006:**
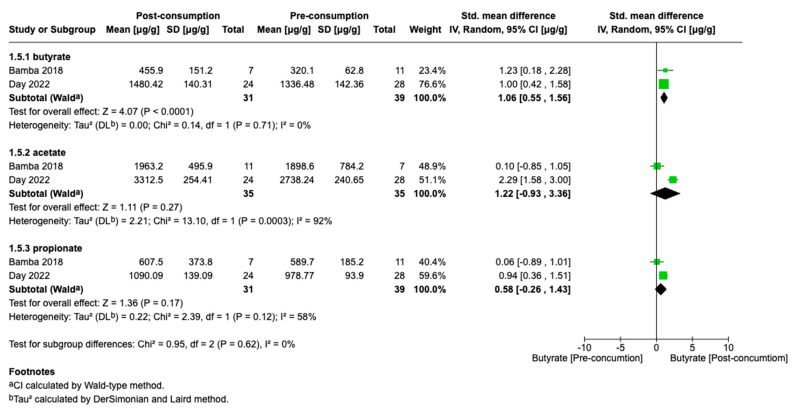
Forest plots of SCFA alterations before and after diet consumption in IBD: 1.5.1 butyrate, 1.5.2 acetate, 1.5.3 propionate (Bamba 2018 [32]; Day 2022 [34]).

**Table 1 biomolecules-15-01017-t001:** Characteristics of the included case–control studies.

Study	Country	Study Type	Type of IBD	Case/ Control	Control			IBD			Unit	Data Format Provided for SCFAs	Methods for Measure SCFAs
								UC/CD					
					N	Median Age, Years	Sex (F/M)	N	Median Age, Years	Sex (F/M)			
Kaczmarczyk, [35]	Poland	Case–control	UC/CD	IBD/control	4	48	2/2	23/8	32/33.5	(10/13)/(3/5)	µg/g	median (Q1–Q3)	CE-UV
Dabek-Drobny, [36]	Poland	Case–control	UC/CD	IBD/control	12	23.5	3/9	26/10	35/28	(9/17)/(5/5)	µg/g	median (Q1–Q3)	CE-UV
Ozturk, [33]	Turkey	Case–control	UC/CD	IBD/control	30	44	13/17	17/17	40/49	(7/10)/(7/10)	mmol/kg	Mean, SD; median (Q1–Q3)	GC-MS
Vich Vila, [30]	The Netherlands	Case–control	UC/CD	IBD/control	255	46.83	141/114	176/239	42.61	254/170	μg/g	Mean, SD	LC-MS-MS
Ambrozkiewicz, [38]	Poland	Case–control	CD	IBD/control	9	32	6/3	15	32	5/10	[ppm]	Mean, SD	GC-MS

CE-UV—capillary electrophoresis with spectrophotometry; GC-MS—gas chromatography–mass spectrometry; LC-MS-MS—liquid chromatography with tandem mass spectrometry.

**Table 2 biomolecules-15-01017-t002:** Characteristics of the included studies.

Study	Country	Study Type	Type of IBD	IBD Diagnosis/Activity	Case/Control	Control			IBD			Unit	Data Format Provided for SCFAs	Methods for Measure SCFAs
									Inactive/Active					
						N	Median Age, Years	Sex (F/M)	N	Median Age, Years	Sex (F/M)			
Kaczmarczyk-2, [31]	Poland	Case–control	UC/CD	Mayo score	IBD/control	6	43.8	4/2	11/18	39.8	(2/9)/(2/12)	µg/g	median (Q1–Q3)	CE-UV
Kaczmarczyk-3, [37]	Poland	Case–control	IBD	Mayo Score/CDAI, colonoscopy, and/or imaging	IBD/control	16	31.7	13/3	22/39	36.8/32.5	22/39	μg/g	median (Q1–Q3)	CE-UV

CE-UV—capillary electrophoresis with spectrophotometry.

**Table 3 biomolecules-15-01017-t003:** Characteristics of the included studies.

Study	Country	Study Type	Type of IBD	Diet	Case/Control	IBD			Unit	Data Format Provided for SCFAs	Methods for Measure SCFAs
						Pre-Consumption/Post-Consumption					
						N	Median Age, Years	Sex (F/M)			
Day, [34]	Australia	prospective, 8 wk, open-label feasibility study	UC	SUlfide-REduction (4-SURE) Diet	W8/W0	28/28	42	15/13	μmol/g	median (Q1–Q3)	GC
Bamba, [32]	Japan	an open-label randomized controlled trial	UC	fermented vegetable beverage	UC before and after diet	11/11	43/58	6/5	µmol/g	median (Q1–Q3)	HPLC

GC—gas chromatography; HPLC—high-performance liquid chromatography.

## Data Availability

No new data were created or analyzed in this study.

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
