# Peer review of "Quantitative Alterations in Short-Chain Fatty Acids in Inflammatory Bowel Disease: A Systematic Review and Meta-Analysis"

_biomolecules, 2025, doi:10.3390/biom15071017_

Round 1
Reviewer 1 Report
Comments and Suggestions for Authors
The systematic review and meta-analysis conducted by the authors highlight a significant finding regarding quantitative alterations in short-chain fatty acids (SCFAs) in patients with chronic inflammatory bowel disease. However, the substantial methodological and clinical heterogeneity among the included studies partially limits the robustness of the conclusions drawn.
Although it is often stated that short-chain fatty acid (SCFA) production is reduced in inflammatory bowel diseases (IBD), such claims frequently lack robust empirical support. This review contributes to clarifying several key aspects of the issue; however, certain relevant points appear to be overlooked and would benefit from further elaboration by the authors.
1) From Table 1, particularly the last column detailing the methods used for SCFA quantification, a marked heterogeneity among the eight analytical techniques employed in the studies included in the meta-analysis becomes evident, with significant differences in analytical sensitivity and specificity. For instance, capillary electrophoresis with UV detection (CE-UV) may be less sensitive than more advanced techniques such as gas chromatography–mass spectrometry (GC-MS), and may be limited in its ability to detect SCFAs present at very low concentrations. While GC-MS offers higher sensitivity and selectivity, it requires expensive instrumentation, advanced technical expertise, and additional steps such as derivatization, which increase analytical complexity and the potential for error. High-performance liquid chromatography (HPLC), on the other hand, may have lower sensitivity compared to GC-MS or LC-MS/MS, and sometimes insufficient resolution for the analysis of complex samples. These methodological aspects should be more clearly discussed in the manuscript, as they may significantly contribute to the high degree of heterogeneity (I²) frequently reported in the results. Finally, it would be of interest—although the limited number of available references may constrain such an approach—to assess separately the results obtained using CE-UV alone, in order to explore the impact of analytical methodology on overall variability.
2) Another relevant aspect not addressed by the authors is the phenomenon of microbial cross-feeding among different bacterial taxa. Many gut bacteria do not produce butyrate directly from dietary fibers but instead synthesize it by utilizing intermediate metabolites such as acetate and propionate. A well-known example is Faecalibacterium prausnitzii, which converts acetate into butyrate, whereas only a relatively limited number of species, such as Clostridium butyricum, are capable of producing butyrate directly from fiber fermentation. Therefore, the observed reduction in acetate levels—particularly pronounced—should be interpreted not only in terms of its primary production, but also in light of its metabolic utilization by the gut microbial community.
In conclusion, the manuscript highlights a potential difference in the production and absorption profiles of short-chain fatty acids (SCFAs) between ulcerative colitis (UC) and Crohn’s disease (CD), and the authors propose several interpretative hypotheses to explain these divergences. While of interest, these hypotheses could be further strengthened and refined through the integration of targeted biochemical and microbiological analyses.
Author Response
Comments 1: [From Table 1, particularly the last column detailing the methods used for SCFA quantification, a marked heterogeneity among the eight analytical techniques employed in the studies included in the meta-analysis becomes evident, with significant differences in analytical sensitivity and specificity. For instance, capillary electrophoresis with UV detection (CE-UV) may be less sensitive than more advanced techniques such as gas chromatography–mass spectrometry (GC-MS), and may be limited in its ability to detect SCFAs present at very low concentrations. While GC-MS offers higher sensitivity and selectivity, it requires expensive instrumentation, advanced technical expertise, and additional steps such as derivatization, which increase analytical complexity and the potential for error. High-performance liquid chromatography (HPLC), on the other hand, may have lower sensitivity compared to GC-MS or LC-MS/MS, and sometimes insufficient resolution for the analysis of complex samples. These methodological aspects should be more clearly discussed in the manuscript, as they may significantly contribute to the high degree of heterogeneity (I²) frequently reported in the results. Finally, it would be of interest—although the limited number of available references may constrain such an approach—to assess separately the results obtained using CE-UV alone, in order to explore the impact of analytical methodology on overall variability.]
|
Response 1: Dear Reviewer, Thank you for your valuable feedback. We sincerely thank the reviewer for this insightful and constructive comment. We agree that the heterogeneity in analytical techniques used for SCFA quantification across the included studies—particularly the variation in sensitivity and specificity—represents an important source of variability that may have contributed to the high I² values observed in our meta-analysis. In response to your suggestion, we have now revised the Discussion section to explicitly address the potential impact of methodological differences on the overall heterogeneity. This clarification has been incorporated into the discussion section and in the lines 402-417.
|
Moreover, In two studies where capillary electrophoresis with UV detection (CE-UV) was used as the analytical method for SCFA quantification in CD and UC, a high degree of heterogeneity was observed. However, this heterogeneity appears to be driven not by methodological differences, but rather by limited sample sizes. The study by Dabek-Drobny, 2022 included 26 participants, whereas the study by Kaczmarczyk, 2021 involved only 7. Due to the small number of studies in our analysis that exclusively used CE-UV, it was not feasible to perform a separate evaluation of results to isolate the impact of analytical methodology on overall variability. Further studies with larger sample sizes are needed to more accurately assess the contribution of analytical techniques to result heterogeneity.
|
Comments 2: [Another relevant aspect not addressed by the authors is the phenomenon of microbial cross-feeding among different bacterial taxa. Many gut bacteria do not produce butyrate directly from dietary fibers but instead synthesize it by utilizing intermediate metabolites such as acetate and propionate. A well-known example is Faecalibacterium prausnitzii, which converts acetate into butyrate, whereas only a relatively limited number of species, such as Clostridium butyricum, are capable of producing butyrate directly from fiber fermentation. Therefore, the observed reduction in acetate levels—particularly pronounced—should be interpreted not only in terms of its primary production, but also in light of its metabolic utilization by the gut microbial community.]
Thank you for this valuable comment. We fully agree that many gut bacteria, including Faecalibacterium prausnitzii, do not produce butyrate directly from dietary fiber, but rather utilize intermediate metabolites such as acetate and propionate in the process of butyrate synthesis. As you correctly pointed out, only a limited number of species—such as Clostridium butyricum—are capable of converting fiber directly into butyrate. In light of this, we have revised the manuscript to emphasize that the observed reduction in acetate levels should be interpreted not only in terms of primary production but also in the context of its metabolic consumption by butyrate-producing bacteria. This clarification has been incorporated into the discussion section and in the lines 427-445.
Thank you again for helping us improve the clarity and scientific rigor of our manuscript.

Reviewer 2 Report
Comments and Suggestions for Authors
Chulenbyeva et al. conducted a meta-analysis to investigate the changes in fecal short-chain fatty acids (SCFAs) and their potential roles in inflammatory bowel disease (IBD). The results are promising but require substantial revisions before publication.
1. In the Introduction section, the authors could elaborate on the differences in treatment strategies for Crohn's Disease (CD) and Ulcerative Colitis (UC).
2. Spelling errors should be corrected, such as "that may exacerbate pathological processe" (line 80) and "tudy data" (line 119).
3. The study assessed alterations in the levels of butyrate, acetate, and propionate in IBD patients. What are the possible changes in other SCFAs?
4. Could the results from colitis animal models be integrated into this study?
5. How do the changes in SCFA levels influence IBD treatment strategies?
6. Figure quality should be improved.
Author Response
Comments 1: [In the Introduction section, the authors could elaborate on the differences in treatment strategies for Crohn's Disease (CD) and Ulcerative Colitis (UC).]
|
Response 1: Dear Reviewer, Thank you for your valuable feedback. We thank you for this helpful suggestion. In response, we have expanded the Introduction section to include a brief overview of the differences in treatment strategies for Crohn’s disease (CD) and ulcerative colitis (UC). Specifically, we now highlight that treatment for UC often begins with aminosalicylates, while CD generally requires earlier use of immunosuppressive or biologic agents due to its transmural and variable presentation. These distinctions are important for understanding disease behavior, which may also be reflected in microbial and metabolic differences explored in our study. The revised text can be found on page 2, lines 61-70. |
|
Comments 2: [Spelling errors should be corrected, such as "that may exacerbate pathological processe" (line 80) and "tudy data" (line 119).]
Response 2:
Thank you for pointing out these typographical errors. We have corrected the misspellings, including “processe” to “processes” (line 80) and “tudy data” to “study data” (line 119), and have thoroughly proofread the manuscript to ensure there are no remaining spelling mistakes. We appreciate your careful review and attention to detail.
Comments 3: [The study assessed alterations in the levels of butyrate, acetate, and propionate in IBD patients. What are the possible changes in other SCFAs?]
Response 3:
Thank you for your thoughtful comment. In our study, we focused on butyrate, acetate, and propionate, as they are the most abundant and well-studied short-chain fatty acids (SCFAs) in the context of gut health and inflammatory bowel disease (IBD). We agree that investigating other SCFAs, such as valerate, isobutyrate, and isovalerate, may provide additional insights into microbial metabolism and host-microbe interactions. However, due to limitations in the available data across included studies, we were unable to consistently extract and analyze information on these less prevalent SCFAs. We have now acknowledged this as a limitation in the revised manuscript and suggested it as a direction for future research.
Comments 4: [Could the results from colitis animal models be integrated into this study?]
Response 4:
Thank you for this insightful suggestion. While we acknowledge the valuable contributions of colitis animal models to understanding the mechanisms underlying IBD, the scope of our study was specifically limited to human subjects. Our aim was to focus exclusively on SCFA alterations observed in clinical human populations to enhance the translational relevance of our findings. We have clarified this point in the revised manuscript to avoid any confusion regarding the study's scope.
Comments 5: [How do the changes in SCFA levels influence IBD treatment strategies?]
Response 5:
Thank you for this thoughtful question. We agree that understanding how SCFA alterations influence treatment strategies is of great importance. In our manuscript, we aimed to address this by highlighting the potential clinical implications of SCFA profiling. Specifically, we suggest that changes in SCFA levels could serve as biomarkers for disease activity and subtype differentiation, and may guide therapeutic modulation through dietary interventions or microbiota-targeted therapies. This has now been clarified and emphasized in the revised Discussion section to better reflect the translational potential of our findings.
Comments 5: [Figure quality should be improved]
Response 5:
Thank you for your helpful feedback. We have revised the figure(s) to improve their resolution and overall quality for better clarity and readability. The updated versions have been included in the revised manuscript.
Thank you again for helping us improve the clarity and scientific rigor of our manuscript.

Round 2
Reviewer 2 Report
Comments and Suggestions for Authors
The authors have addressed all my concerns.